# Measuring Single-Cell Calcium Dynamics Using a Myofilament-Localized Optical Biosensor in hiPSC-CMs Derived from DCM Patients

**DOI:** 10.3390/cells12212526

**Published:** 2023-10-26

**Authors:** Cara Hawey, Kyla Bourque, Karima Alim, Ida Derish, Elise Rody, Kashif Khan, Natalie Gendron, Renzo Cecere, Nadia Giannetti, Terence E. Hébert

**Affiliations:** 1Department of Pharmacology and Therapeutics, McGill University, Montréal, QC H3G 1Y6, Canada; cara.hawey@mail.mcgill.ca (C.H.); kyla.bourque@mail.mcgill.ca (K.B.); karima.alim@mail.mcgill.ca (K.A.); 2Research Institute, McGull University Hospital Centre, 1001 Decarie Blvd, Montréal, QC H4A 3J1, Canada; ida.derish@mail.mcgill.ca (I.D.); elise.rody@muhc.mcgill.ca (E.R.); kashif.khan2@mail.mcgill.ca (K.K.); natalie.gendron@muhc.mcgill.ca (N.G.); renzo.cecere@mcgill.ca (R.C.); nadia.giannetti@mcgill.ca (N.G.)

**Keywords:** calcium handling, cardiomyocytes, dilated cardiomyopathy, cellular signaling

## Abstract

Synchronized contractions of cardiomyocytes within the heart are tightly coupled to electrical stimulation known as excitation-contraction coupling. Calcium plays a key role in this process and dysregulated calcium handling can significantly impair cardiac function and lead to the development of cardiomyopathies and heart failure. Here, we describe a method and analytical technique to study myofilament-localized calcium signaling using the intensity-based fluorescent biosensor, RGECO-TnT. Dilated cardiomyopathy is a heart muscle disease that negatively impacts the heart’s contractile function following dilatation of the left ventricle. We demonstrate how this biosensor can be used to characterize 2D hiPSC-CMs monolayers generated from a healthy control subject compared to two patients diagnosed with dilated cardiomyopathy. Lastly, we provide a step-by-step guide for single-cell data analysis and describe a custom *Transient Analysis* application, specifically designed to quantify features of calcium transients. All in all, we explain how this analytical approach can be applied to phenotype hiPSC-CM behaviours and stratify patient responses to identify perturbations in calcium signaling.

## 1. Introduction

Dilated cardiomyopathy (DCM) is a multifaceted disease that impacts the structure and function of the heart muscle. The maladaptive dilatation of the left ventricle makes DCM a leading indication for heart transplantation and has a 50% mortality rate within 5 years of diagnosis [1]. DCM is diagnosed because of numerous etiologies including mutations in genes encoding sarcomeric proteins, viral infections, excessive alcohol consumption, and exposure to toxins or chemotherapeutic drugs, among other causes [2]. DCM-causing mutations have been identified in genes encoding proteins that regulate cardiac contractions, such as members of the troponin complex (cTnT/I/C), myosin heavy chain (MYH6 and MYH7), and myosin light chain (actin) [3,4]. Human induced pluripotent stem cell-derived cardiomyocytes (hiPSC-CMs) from patients carrying mutations in these genes have shown impairments in contractility and dysfunctional calcium homeostasis [1,5,6,7]. Calcium handling is a tightly regulated process involved in the cardiac contraction cycle and is associated with disease pathogenesis when dysregulated [8,9]. Cardiomyocytes contract via a mechanism known as “excitation-contraction coupling”. This process is initiated by an action potential that depolarizes the plasma membrane of cardiomyocytes, which triggers the opening of L-type calcium channels (LTCCs) that allow the influx of extracellular calcium (Figure 1) [10]. Calcium is also released from intracellular stores in the sarcoplasmic reticulum (SR) through ryanodine receptors (RyR2, in fact) in a process known as “calcium-induced calcium release” [10]. Calcium then binds to sarcomeres, the contractile units of the cardiomyocyte. Sarcomeres are composed of interlacing patterns of thin (actin) and thick (myosin) myofilaments as well as the troponin complex (T/C/I) and tropomyosin, among other proteins. Calcium binds the troponin complex which induces a conformational change in tropomyosin that allows actin to bind myosin, resulting in contraction of the cardiomyocyte (Figure 1) [11]. When calcium is released from troponin, the contraction cycle ends, and calcium returns to the SR via the sarco-endoplasmic reticulum Ca^2+^-ATPase (SERCA2a) or exits the cell via the sodium-calcium exchanger (NCX) (Figure 1) [12]. It is essential for this process of excitation-contraction coupling to be tightly regulated as dysregulation can lead to disease. For example, phospholamban (PLN) is a regulatory protein that inhibits SERCA2a to control calcium reuptake into the SR. When phosphorylated by protein kinase A (PKA), PLN releases SERCA2a and allows for calcium reuptake. This release can be hindered by a mutation in PLN’s binding site for PKA (PLN^R9C^), resulting in constitutive SERCA2a inhibition and prolonged calcium transient decay [13]. Individuals with PLN^R9C^ develop dilated cardiomyopathy with decreased contractile function that leads to severe heart failure and ultimately the need for a heart transplant [13].

Calcium handling is compromised in DCM, as indicated by the altered expression and function of calcium handling proteins, altered sensitivity to calcium at the myofilaments, reduced contractile function, and decreased calcium transient peak height during systole and calcium reuptake during diastole [14]. To study whether calcium handling is perturbed due to disease, various features of the calcium cycle can be measured on a patient-specific basis using cardiomyocytes derived from human iPSCs (iPSC-CMs). The latter have been successfully used to model numerous cardiovascular diseases in a dish, including several reports of DCM [1,7,8,9]. Once differentiated into cardiomyocytes, these cells have been shown to recapitulate many disease features and provide a very powerful tool for personalized disease modeling and precision medicine.

### 1.1. Methods to Study Ca^2+^ Signaling in Cardiomyocytes

Various methods are available to measure intracellular calcium in hiPSC-CMs and other cell types (shown schematically in Figure 2). The most commonly used methods are intracellular calcium-sensitive dyes and genetically encoded calcium indicators (GECIs). Calcium-sensitive dyes such as Fluo-4 and Fura-2 fluoresce in the presence of calcium. These compounds are bound to an acetomethoxy (AM) ester that allows them to cross cell membranes to enter the cytosol where they are cleaved by esterases, trapping the released Fluo-4 and Fura-2 in the cell. Fluorescence intensity of these dyes correlates with the concentration of calcium inside of the cell. Dyes can be single-wavelength (e.g., Fluo-4) or double-wavelength (e.g., Fura-2) indicators. Single-wavelength calcium indicators have higher fluorescence intensities compared to double; however, double-wavelength calcium indicators give ratiometric readouts that allow for quantitative results [15,16].

Genetically encoded calcium indicators, or GECIs, are biosensors that can be expressed in mammalian cells and fluoresce in response to calcium binding/activity. When transfected or transduced into cells, calcium can bind the calcium-binding domain of the GECI to induce a conformational change that results in the activation of the attached fluorescent protein(s). Among the most widely used GECIs are the GCaMP family of calcium biosensors. GCaMP is composed of a circularly permutated enhanced GFP (cpEGFP), calmodulin, and calmodulin target sequence M13 [17]. Upon calcium binding to calmodulin, a conformational change is induced in cpEGFP, resulting in increased fluorescence intensity. Modifications to the parent GCaMP biosensor have been made to improve fluorescence intensity, sensitivity, and dynamic range for calcium detection [18,19,20,21]. In addition to GCaMP, the genetically encoded calcium indicator for optical imaging (GECO)’s series of sensors can be used to detect intracellular calcium. While GCaMPs fluoresce in the green spectrum, GECO biosensors are also available in the red spectrum (RGECO). The longer wavelength reduces phototoxicity and provides greater tissue penetration, making these GECIs useful in long-term experiments and in 3-dimensional organoid models [22].

Biosensors localizing to subcellular compartments, such as the mitochondria, plasma membrane, nucleus, endo/sarcoplasmic reticulum, and myofilament, are useful in providing a more detailed picture of intracellular calcium handling [23,24,25]. To visualize calcium handling at the myofilament, troponin T (TnT) has been conjugated to RGECO, generating RGECO-TnT [25]. In the first report of this biosensor, the authors demonstrated that myofilament and whole-cell calcium dynamics can differ [25]. For instance, cells treated with ATPase inhibitor MYK-461 showed altered calcium binding, release, and signal amplitude in myofilament-localized calcium handling; however, these results were not reported in whole-cell calcium transients [25]. Furthermore, calcium sensitizer levosimendan induced a larger effect size at the myofilament versus whole-cell level [25]. Using RGECO-TnT, calcium handling at the myofilament microdomain can be measured and used to elucidate disease phenotypes when combined with hiPSC-CMs derived from DCM patients, as described below.

### 1.2. Analyzing Calcium Transients Derived from hiPSC-CM Monolayers

There are several parameters that may be extracted from calcium traces to detect abnormal transients and associated disease phenotypes. These parameters can be compared between patients and control subjects to understand how the disease alters calcium handling, as well as to evaluate drug responses to identify novel therapies or screen for potential cardiotoxicities. Some features of transients that can be evaluated include transient frequency, transient amplitude, transient duration, upstroke velocity, area under the curve, decay tau, and time taken to reach 50% or 90% baseline (Figure 3). Various programs are available to quantify and analyze calcium transients; however, large single-cell datasets present a challenge in the analysis pipeline. To address this, we discuss a new program available to analyze calcium transients and use hiPSC-CMs derived from DCM patients as a model system.

There are several calcium analysis programs currently available. Proprietary analysis programs such as the pCLAMP 11 Software Suite (Molecular Devices) and IonWizard v. 7.3 (IonOptix) are available but require the purchasing of a licence. pCLAMP11 (RRID:SCR_011323) is typically used with patch clamp amplifiers to analyze electrophysiological data while IonWizard (RRID:SCR_021764) is part of a microscopy system designed to measure calcium handling and contractility in cardiomyocytes. To circumvent the need for proprietary programs, an Excel-based program was developed to process raw signal and analyze results in the same program as well as to exponentially fit the curves and determine decay tau [26]. In recent years, research groups have been developing open-access programs that are more user-friendly, compute more features of Ca^2+^ transients, and are capable of handling large datasets. One such program is the CardIAP web app that allows users to analyze calcium transients from confocal line-scan images to evaluate transient amplitude, time to peak, time to half peak, and decay tau [27]. In a similar way, AnomalyExplorer is an interactive software tool that identifies anomalies in calcium transients [28]. The software can identify normal peaks, abnormal plateaus, oscillation, irregular phases, middle peaks, double peaks, and low peaks in calcium transients and expresses results as the proportion of cells that fall within each group. However, AnomalyExplorer does not quantify features of calcium transients such as decay tau, transient amplitude, and others mentioned above.

Further, CalTrack is a MatLab-based program that is used to assess whole-cell calcium transients in cardiomyocytes [29]. This in-depth program can be used to extract fluorescent traces from videos containing single or multiple cells, graph the traces, and perform analysis to quantify various features of calcium transients such as decay tau, transient duration, beat rate, and amplitude, among other features. CalTrack provides a platform to rapidly process large datasets from captured videos or extracted intensities; however, in our hands, it was unable to accurately identify individual cells expressing RGECO-TnT, since cell borders were not always well defined. It is also challenging for individuals not familiar with the MatLab language. Below, we describe a Calcium Transient Analysis application developed in collaboration with OriginLab to analyze calcium transients measured in hiPSC-CMs derived from control subjects and DCM patients. This analysis pipeline begins by extracting fluorescent intensities using ImageJ, a free software available online (https://imagej.net/ij/index.html). Subsequently, intensities are imported into OriginPro 2022 (RRID:SCR_014212; OriginLab) where the Calcium Transient Analysis application is integrated. The application automatically computes 11 features of calcium transients (Figure 3) that can be graphed and statistically analyzed in OriginPro 2022 or other statistical analysis and graphing software.

## 2. Materials and Methods

### 2.1. Transduction of hiPSC-CMs with AAV2/6-RGECO-TnT

The plasmid pShuttle CMV RGECO-TnT (Addgene plasmid #124643; http://n2t.net/addgene:124643; RRID:Addgene_126463) was cut using Acc65I and HindIII and packaged in an adeno-associated virus (AAV-CAG) backbone. AAV2/6-RGECO-TnT was produced by the Neurophotonics Platform Viral Vector Core at Laval University, Québec.

Note: For stability, keep AAV2/6-RGECO-TnT on ice. Use low retention tubes and filter tips when handling the virus. For long-term storage, it is recommended to store AAVs at −80 °C and aliquot into small volumes to avoid multiple freeze-thaw cycles.

Refresh media in 96-well plate with 70 μL of warmed RPMI + B27.Prepare a low retention 1.5 mL tube with enough media to add 10 μL per well containing cells to be transduced (i.e., 100 μL for 9 wells, account for pipette error).Thaw an aliquot of AAV2/6-RGECO-TnT on ice.Determine the volume of virus required to infect the cells with a MOI of 5000 for all wells and transfer to the prepared 1.5 mL tube containing media using a filter tip. Mix by inverting the Eppendorf gently 3–4 times and return the tube to the ice.Immediately add 10 μL of virus dilution per well.Incubate the dish with the virus for 5 days, replacing the media with 100 μL RPMI + B27 3 days after transduction.On the day of the experiment, replace media with RPMI1640 (no phenol red) supplemented with B27 (with insulin).

### 2.2. Live Cell Ca^2+^ Imaging of hiPSC-CM Monolayers

Calcium imaging was performed using the Zeiss Axio Observer (Dorval, Québec, Canada) fully-automated inverted microscope and Zen Blue software v3.3 (Appendix A). To begin, it is recommended to turn on the CO_2_ supply, temperature control, and gas-mixing unit 10 min before use to ensure a physiological environment for the cells during imaging. Importantly, this system should be turned on in a particular order to confirm that all components accurately recognize and communicate with each other. If this order is not followed, the Zen Blue software v3.3 will display error messages. To avoid any errors, turn on the system in the following order: (1) main power (Power Supply 232), (2) stage controller (SMC 2009), (3) definite focus (Focus Controller 2), (4) microscope, (5) X-cite 120 LED, then (6) Zen Blue software. Once the microscope is responsive, the following instructions can be followed:Ensure a Zeiss Filter Set 14 is installed (510–560 nm excitation, 590 nm emission, and 580 dichroic mirror).

Note: The original RGECO-TnT publication used a filter set with a 520/25 nm excitation, 620/60 nm emission and a 565 dichroic mirror [25].

Enable high-speed imaging according to the manufacturer’s recommendation. Some examples to improve speed on the Axio Observer include using the GPO 0 trigger cable, on Zen Blue enable “Trigger Out” for live, and set shutter open delay to 0.00 ms. Binning may be used to improve fluorescent signal; for example, binning can be set to 3 × 3.In the software, under “Imaging Setup” select the appropriate filter set and set the objective lens to 20X/0.8 (ZEISS 20x PLAN APOCHROMAT, NA 0.8).Under time series, set to record for 15 s and enable “Camera streaming if possible” to improve speed of acquisition.Using “Imaging Setup” on the computer software, or the touch screen panel on the microscope, ensure the correct objective and filter set are selected. Use TL Illumination to see the cells under brightfield and RL Illumination to see the cells under fluorescent lighting. Do not have TL Illumination on during fluorescent recording.Allow hiPSC-CMs to stabilize in the chambered plate holder for a minimum of 15 min before acquiring measurements.To minimize the amount of time cells are exposed to fluorescence, first find cells under TL Illumination. Use the stage controller joystick to move the plate and focus on the cells. Once cells are found, using Zen Blue software under the Acquisition tab, select “Live” to see a live stream of the cells. Ensure the cells are in focus before selecting “Start Experiment”.Image stacks are recorded as .czi files that can be opened in ImageJ with the Bio-Formats plug-in.

**Note:** Microscopy must be conducted in the dark to minimize background. Turn off lights in the microscopy room for best results.

### 2.3. Image Analysis

Analysis was performed using ImageJ (free download at: https://imagej.nih.gov/ij/download/) and a custom-made application in OriginPro 2022. Briefly, images (saved as .czi files) were opened in ImageJ and cells were selected manually. Using ImageJ, the fluorescence intensities of each individually selected cell were extracted and imported to OriginPro 2022 where the background fluorescence was normalized. The Calcium Transient Analysis application was run to determine features of calcium transients. Statistical analyses and graphing were performed in OriginPro 2022. Readers can refer to Figure 3 for a graphical overview of the workflow.

#### 2.3.1. Extracting Fluorescence Intensities of hiPSC-CM Monolayers in Image J

To extract fluorescence intensities in ImageJ, please follow the steps below:Open files from file explorer or through ImageJ (File > Open…).Once all images have been opened, under the “Image tab”, select “Adjust > Brightness/Contrast”. Then select “Auto” followed by “Set” and select “Propagate to all other open images” to adjust the brightness and contrast of all images simultaneously.Using the “Polygon” or “Freehand” selections tools, trace individual cells and press the “T” key on the keyboard following every cell to add the selected cell as a new region of interest, or ROI, to the ROI manager. Remember to select the background with no cells as the final ROI, as this is required for normalization purposes.In the ROI manager, select “More” >> “Multi Measure”. Ensure “Measure all slices” and “One row per slice” are selected. This will generate a table containing the fluorescent values for all ROIs selected in the image. The information found in this table can be transferred to OriginPro 2022 for further analysis.To perform calculations, first select the green lock in the top corner of A (X) and select “Recalculation Mode” >> “None” and select “OK” in the dialog box.Under column A, transform frame number to time by typing A * (time of recording/number of frames) in “F (x) = X (i.e., A * (15/156)).Normalize cell fluorescence intensities to background intensity by writing in F (x) = (cell column or “this” – background column)/background column (i.e., (B-F)/F or (this – F)/F).

Note: Use the notation “this” to apply the same formula to multiple columns. “this” is the variable that refers to the dataset within a given column.

#### 2.3.2. Calcium Transient Analysis of hiPSC-CM Monolayers Using OriginPro

In OriginPro 2022, under “Add Apps“ search for “Calcium Transient Analysis” to download the application. The application may also be found at the following link: https://www.originlab.com/FileExchange/details.aspx?v=0&fid=900). The first time the program is run, users need to select “Install Python Packages”. These packages only need to be downloaded once.

Once the cell fluorescence intensities have been normalized to the background using ImageJ, as described above, features of calcium transients can be computed. To do this, select “Calcium Transient Analysis” under “Apps” to compute features of calcium transients. A dialog box will then appear with options for “windows size”, “tolerance forward”, and “tolerance backwards”. These parameters can be changed to more accurately detect the beginning and end of the transient, as well as the number of peaks. The application has built-in capacity to calculate 11 features of calcium transients: time to peak, time to 90% baseline, time to 50% baseline, time between maximum, transient duration, inter-event interval, average peak height (transient amplitude), average peak area (area under the curve), upstroke velocity, decay tau, and transient frequency (beat rate); refer to Figure 4 for a visual depiction and corresponding definitions. Once the data are imported, a tab is generated for each individual cell, as well as a summary tab containing information for all cells and a transposed summary tab Users can rename this workbook as they see fit. The application also generates a worksheet containing all the single-cell transients that were selected in ImageJ, in a graphed format. Once the single-cell transients have been generated, it is recommended to review each cell tab to ensure the key points of each transient (peak, beginning of the transient, and end of the transient) were accurately detected. If they were not accurately detected, refer to Table 1 for troubleshooting. Once all points of the transient are accurately recognized, select “Calculate” to update the output and summary tabs. Create a new workbook to compile data for graphing and statistical analysis, as desired. For example, a workbook could contain separate tabs for each feature of transients to compare different patients or treatment groups.

#### 2.3.3. Statistical Analysis

Statistical analysis and graphing can be conducted using OriginPro2022′s Paired Comparisons application. To do this, set “Input Data From” to “Raw” and select data columns by selecting the black triangle and selecting the columns to be compared and analyzed. Select “Display” options according to desired output and the appropriate mean comparison methods for the dataset. A graph and summary workbook will be generated with statistical analysis results, which were compared using a one-way ANOVA with a Bonferroni correction. To compile multiple statistical results sheets as tabs under one workbook, the output location can be changed under plot data to select an existing workbook and new sheet.

## 3. Results

Representative calcium transients from hiPSC-CMs derived from a female control subject HID041004, male idiopathic DCM HID041020, and female doxorubicin-induced DCM patient HID041021 are depicted in Figure 5 and Figure 6, respectively. By visualizing single-cell myofilament-localized Ca^2+^ transients, we were able to observe heterogeneity within the overall population of cells sampled. For instance, cells derived from male patient HID041020 displayed uniform transients that did not differ greatly from cell to cell. Interestingly, hiPSC-CMs derived from a female control HID041004 and a female patient HID041021 exhibited more heterogeneity and cell to cell variance as well as irregular transients. Generation and characterization of stem cell lines are described in Supplemental Methods and shown in Appendix A. Subject characteristics are described in Appendix A.

To determine features of calcium transients, values were computed for all transients produced by each cell and averaged to give one value per cell for all features (Figure 7). Differences between patients as well as differences between the patients and control were observed. For example, an increased area under the curve (Figure 7A) was observed in control cells HID041004 versus both DCM patients. The increased transient frequency (Figure 7J) as well as the decreased time to peak (Figure 7F) and time to 90% baseline (Figure 7E) observed in hiPSC-CMs derived from patient HID041020 indicate shorter and more frequent beats compared to the control HID041004 and HID041021. Transient duration (Figure 7B), decay tau (Figure 7G), time between peaks (Figure 7C), time to 50% baseline (Figure 7D), upstroke velocity (Figure 7H), and transient amplitude (Figure 7I) demonstrated statistically significant differences between all hiPSC-CM lines with HID041020 showing the shortest, most frequent, and most rapid transients. These differences between cell lines can be used to gain better insight to mechanisms underlying the disease and to identify potential therapeutic targets [8].

## 4. Discussion

We described a pipeline to measure and analyze myofilament-localized single-cell calcium handling in hiPSC-CMs derived from two DCM patients and a healthy volunteer. In cases where mutations causing DCM are known, it makes more sense to create isogenic control lines using CRISPR/Cas9. Using this pipeline, we were able to detect differences in calcium handling between different cell lines. All features except decay tau showed statistically significant differences in calcium handling between the three cell lines evaluated. Differences between both DCM patients and the healthy control were found in area under the curve, transient amplitude, and transient duration (Figure 7). These results suggest calcium transients in DCM patients are smaller than those seen in the healthy control. Transient frequency, transient duration, time to 50% and 90% baseline, upstroke velocity, time between peaks, and time to peak differed between the two DCM patients, with the hiPSC-CMs derived from the male DCM patient showing faster and shorter transients than hiPSC-CMs derived from a female DCM patient. These results indicate that this pipeline is capable of measuring differences in myofilament-localized single-cell calcium handling on a patient-to-patient and patient-to-control basis. Previous studies have reported differences in calcium handling when comparing hiPSC-CMs with DCM-causing mutations versus controls. It has been reported that when comparing hiPSC-CMs from a DCM patient with a healthy family member, a slower decay tau and delayed time to peak were observed in DCM hiPSC-CMs [30]. Delayed time to peak was also observed in hiPSC-CMs that were CRISPR edited to express a DCM-causing mutation as well as in hiPSC-CMs derived from a DCM patient with a mutation in the same gene (RBM20) [7]. These hiPSC-CMs derived from the patient with RBM20 DCM also showed increased transient amplitude, decay tau, duration of transient, and area under the curve, as well as a decreased transient frequency and no change in upstroke velocity [31]. While this study showed increased transient amplitude, several others reported decreased amplitude in models of DCM [9,32,33]. Differences are to be expected between DCM patients, depending on their etiology of disease and, in the case of genetic DCM, the severity/penetrance of the mutations. These features can be used to gain insight on how dysregulated calcium handling may play a role in disease pathogenesis, progression, or therapeutic responses. For instance, features that measure the decaying arm of the transient (i.e., decay tau, time to baseline) can show delayed calcium removal from the cytosol, which is hypothesized to predispose cardiomyocytes to abnormal electrical activity and, ultimately, arrhythmias and sudden cardiac death [30]. Abnormal transient duration can also be indicative of defective calcium reuptake [31], and decreased calcium transient amplitude can indicate decreased calcium stores in the sarcoplasmic reticulum, which may result from faulty calcium store regulators: RYR2 and SERCA [12,34]. Decreased calcium amplitude can also result from dysregulated binding of calcium to the myofilaments, which may result from mutations in sarcomeric proteins [6,30].

Overall, the Transient Analysis application in OriginPro 2022 offers a platform to quickly compute several features of calcium transients. Fluorescence intensities can be extracted from any image file compatible with the Bio-Formats plug-in of ImageJ and can subsequently be imported into the OriginPro 2022 software. Any necessary calculations or normalizations can be performed in the software before running the Transient Analysis application, for example, normalizing fluorescence intensity to background. Calcium transient graphs for all cells can be quickly verified to ensure accuracy of points indicating beginning, end, and peak of transient and can be adjusted as required. Features of calcium transients can be compiled into separate work sheets to generate graphs using the Paired Comparisons application in OriginPro2022, or any other graphing software. ImageJ is a free software and OriginLab offers a free trial; however, the full version is available for purchase. A student version is available for $69 USD (as of 2023), which is less expensive than other commercially available graphing and statistical analysis software. This pipeline could be paired with calcium transient analysis software AnomalyExplorer, described above, to provide detailed information on features of calcium transients as well as an analysis of the proportion of cells with abnormal transient profiles [28].

While manually selecting cells poses a significant user burden, there is currently no program that can compare to the ability of users to differentiate individual cells expressing RGECO-TnT. The linear nature of myofilaments makes it difficult to accurately identify cell borders, resulting in programs misrecognizing cells; for example, programs will recognize multiple cells within a single cell. To improve upon differentiating individual cells, a membrane marker such as wheat germ agglutinin (WGA) could be used to assist computer programs in accurately identifying cells. A reduction in the number of cardiomyocytes seeded may also represent an alternative. While this pipeline was designed to evaluate myofilament-localized calcium handling, it can also be applied to evaluate whole-cell calcium handling and is compatible with optical voltage indicators. This assay could be further expanded upon to provide more information on the cells and link phenotypes. For example, since RGECO-TnT localizes to the myofilament, sarcomere organization may also be assessed. In this vein, a method could be developed to link a single-cell calcium transient to the degree of organization providing a more in-depth picture of the overall health status of the cell imaged. Other phenotypes that could be evaluated include cell circularity, length, width, length to width ratio, and cell area as well as other subcellular components such as number and distribution of mitochondria. Cell-type-specific antibodies or promoters can also be used to identify atrial, ventricular, and nodal hiPSC-CMs to inform potential differences in lineage-specific calcium handling patterns. Further experiments can investigate sex differences and evaluate response profiles to insults or therapeutic treatment. Long-term and repeated measure experiments can relate calcium handling in hiPSC-CMs back to the clinical phenotype to investigate disease mechanisms and propose new therapeutic strategies.

## Figures and Tables

**Figure 1 cells-12-02526-f001:**
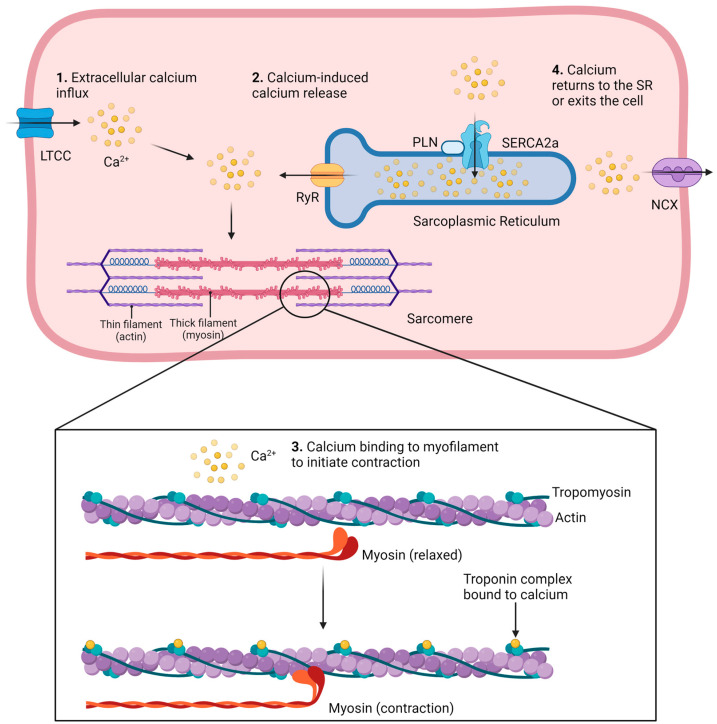
The contraction cycle of cardiomyocytes. Calcium enters the cell to induce a large-scale release of intracellular calcium from the sarcoplasmic reticulum. Calcium then binds to the troponin complex on the myofilament to initiate the contraction. The contraction ceases when calcium returns to intracellular stores or exits the cell.

**Figure 2 cells-12-02526-f002:**
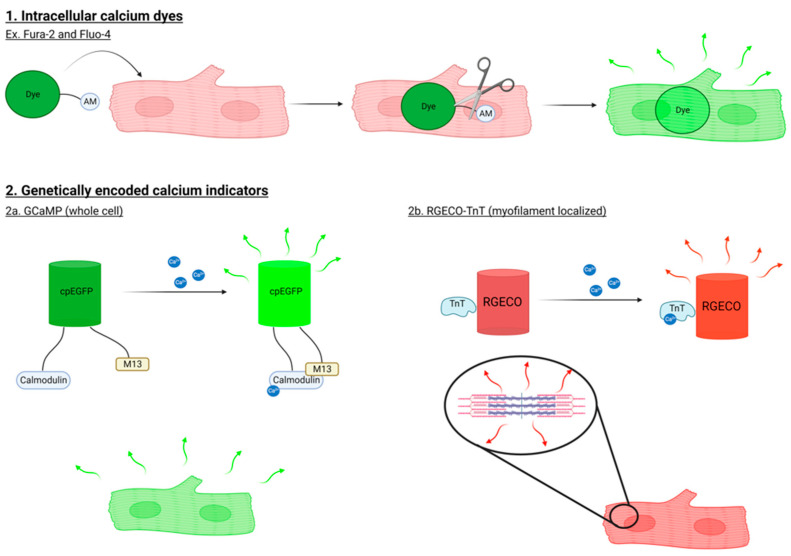
Methods to assess calcium handling in cardiomyocytes. Commonly used methods to study calcium signaling in cardiomyocytes include (**1**) calcium-sensitive dyes and (**2**) genetically encoded calcium indicators (GECIs). (**1**) Calcium-sensitive dyes such as Fluo-4 and Fura-2 enter cells and their acetomethoxy (AM) ester moiety is cleaved, allowing the compounds to fluoresce in response to calcium. (**2**) GECIs are expressed by cells. (**2a**) GCaMP is composed of circularly permutated enhanced GFP (cpEGFP), calcium-binding domain calmodulin, and calmodulin target sequence M13. Calcium binding to calmodulin increases fluorescence intensity. (**2b**) RGECO-TnT is composed of the calcium indicator RGECO and troponin T to localize the GECI to the myofilament in order to assess calcium signalling at the site of contraction.

**Figure 3 cells-12-02526-f003:**
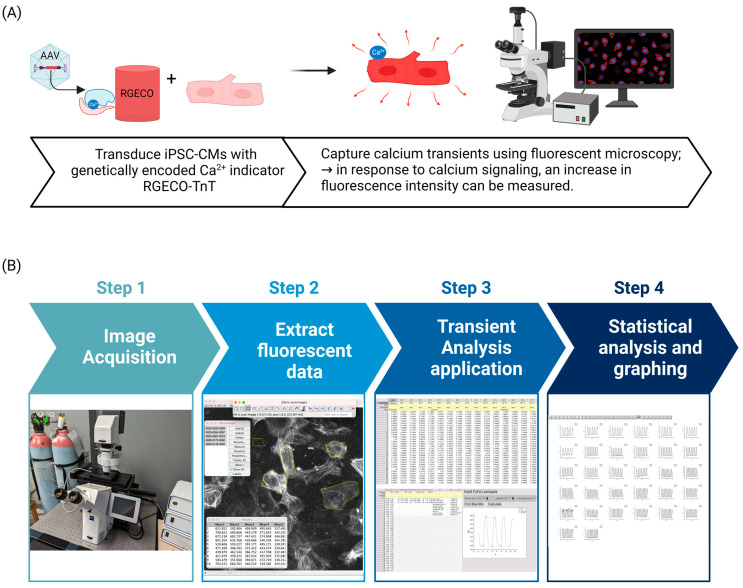
Workflow for imaging and analysis of Ca^2+^ transient in hiPSC-CMs. (**A**) Once hiPSC-CMs are beating in culture, they can be transduced with AAV to deliver the RGECO-TnT biosensor. (**B**) With sufficient biosensor expression, hiPSC-CMs can be imaged. The fluorescence intensity of single cells can then be extracted using ImageJ software. These data can be imported into OriginPro software which can compute 11 features of calcium transients along with graphing and statistical analysis features.

**Figure 4 cells-12-02526-f004:**
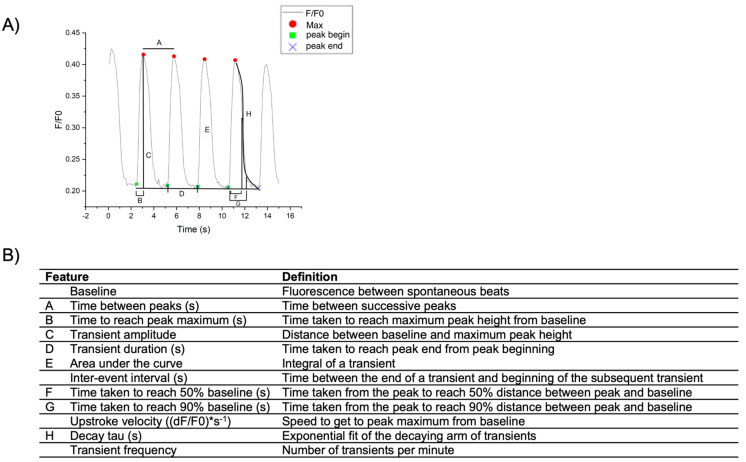
Features of a calcium transient. (**A**) The custom calcium transient analysis application described in the text is capable of computing 11 features of a calcium transient, as depicted. Features of calcium transients include (A) time between peaks, (B) time taken to reach peak maximum, (C) transient amplitude, (D) transient duration, (E) area under the curve, (F) time taken to reach 50% baseline, (G) time taken to reach 90% baseline, and (H) decay tau. Features computed but not shown are upstroke velocity, inter-event interval, and transient frequency. (**B**) Descriptions of each feature that can be extracted from OriginPro.

**Figure 5 cells-12-02526-f005:**
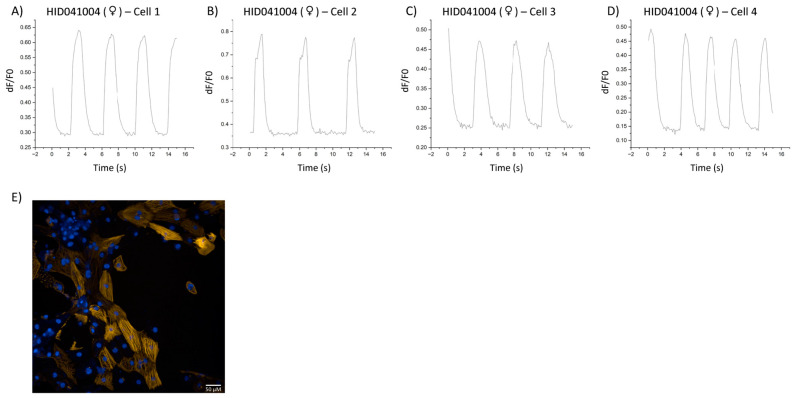
Calcium transients of hiPSC-CMs derived from control HID041004. (**A**–**D**) Single-cell calcium transients from iPSC-CMs derived from a female healthy control (HID041004) visualized using graphs generated in the Calcium Transient Analysis application in OriginPro 2022. (**E**) Microscopy image illustrating the expression of RGECO-TnT in HID041004.

**Figure 6 cells-12-02526-f006:**
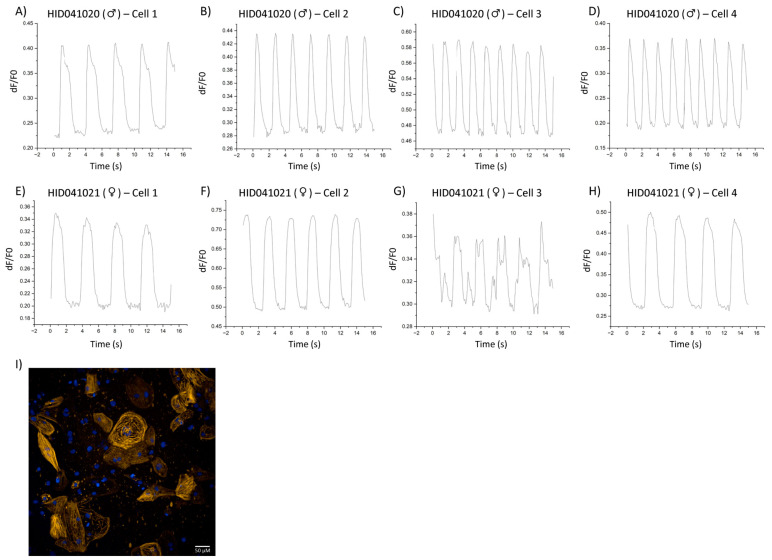
Calcium transients of hiPSC-CMs derived from DCM patients HID041020 and HID041021. (**A**–**D**) Single-cell calcium transients from hiPSC-CMs derived from a male DCM patient (HID041020) visualized using the Calcium Transient Analysis application in OriginPro 2022. (**E**–**H**) Single-cell calcium transients from iPSC-CMs derived from a female DCM patient (HID041021) visualized using the Calcium Transient Analysis application in OriginPro 2022. (**I**) Microscopy image illustrating the expression of RGECO-TnT in HID041020.

**Figure 7 cells-12-02526-f007:**
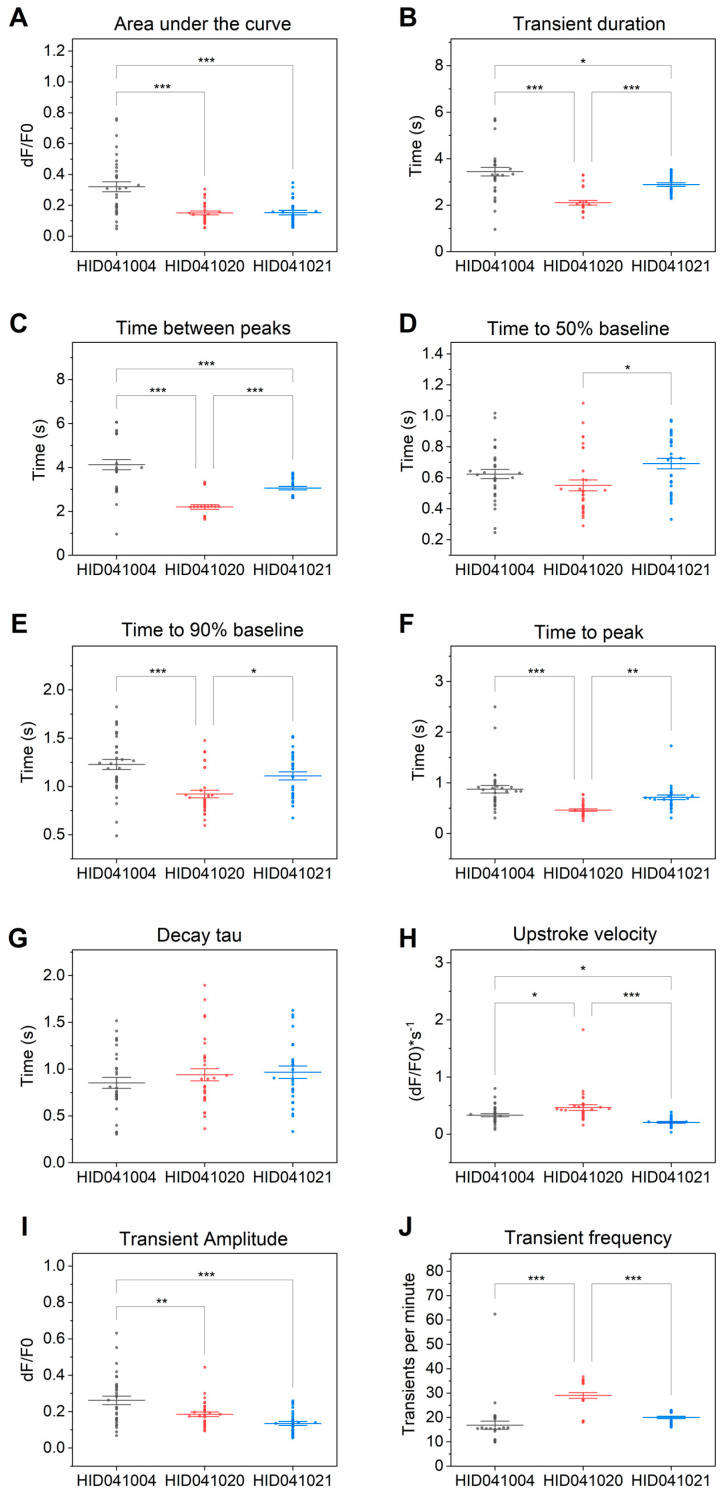
Comparisons of averaged calcium transients in DCM patients vs. control hiPSC-CMs. Features of calcium transients computed using the Calcium Transient Analysis application in OriginPro 2022. Comparisons were made between hiPSC-CMs derived from a female healthy control (HID041004), male DCM patient (HID041020), and female DCM patient (HID041021). Features investigated were (**A**) area under the curve, (**B**) transient duration, (**C**) time between peaks, (**D**) time to 50% baseline, (**E**) time to 90% baseline, (**F**) time to peak, (**G**) decay tau, (**H**) upstroke velocity, (**I**) transient amplitude, and (**J**) transient frequency. Multiple comparisons were made using Bonferroni correction. *n* = 29–33 cells from four independent replicates. * *p* < 0.05, ** *p* < 0.01, *** *p* < 0.001.

**Table 1 cells-12-02526-t001:** Troubleshooting single-cell transients in OriginPro2022.

**If the number of peaks is not accurately recognized.**	Lower Window Size to recognize more peaks and raise Window Size to recognize less peaks
**If the beginning of the transient is not accurately recognized.**	Lower Tol_peakstart to shift green indicators to the left and raise Tol_peakstart to shift green indicators to the right
**If the end of the transient is not accurately recognized.**	Lower Tol_peakend to shift blue x indicators to the left and raise Tol_peakend to shift blue x indicators to the right. If the final transient is incomplete, delete the last point from the spreadsheet.
**If one indicator is out of place while the others are accurate.**	Select on the indicator, select Data at the top of the screen, and select Move Data Points. Select “Yes” in the dialog box. Select and drag the indicator to the desired position then return to the workbook.
**For transients that are noisy** (**i.e., poor signal intensity during acquisition**)**.**	A smoothing filter may be applied. From the tab with the transient to be smoothed, use the top toolbar to select “Analysis” > “Signal Processing” > “Smooth”. Open a new dialog window. Set recalculate to “Manual” and select column B (Y1). Use the desired smoothing method, for example FFT Filter. For output select <input>: XY = input XY to replace the original data with the smooth filtered data.

**NOTE:** Select “Find Max/Min” after each adjustment to see updated results. If required, move individual points at the end and do not select “Find Max/Min” once points are in the correct place as this will reset their positions. Window Size, Tol_peakstart, and Tol_peakend can be changed once the Transient Analysis application is launched if required changes are anticipated. For instance, if a higher transient frequency is anticipated, Window Size can be lowered to proactively apply this change to all cells.

## Data Availability

Data are available upon request.

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
