# Peer review of "Measuring Single-Cell Calcium Dynamics Using a Myofilament-Localized Optical Biosensor in hiPSC-CMs Derived from DCM Patients"

_cells, 2023, doi:10.3390/cells12212526_

Round 1

Reviewer 1 Report

Comments and Suggestions for Authors

In the review article 'Measuring Single-Cell Calcium Dynamics Using a Myofilament-Localized Optical Biosensor in hiPSC-CMs Derived from DCM Patients', submitted by Cara Hawey and coworkers to Cells, the authors reviewed the literature about calcium measurements in hiPSC-derived cardiomyocytes. 

The manuscript addresses an important topic and is well written. It has impact for many readers in cardiovascular genetics. However, some points can be improved in a revised manuscript:

1.)  Please add an OMIM identifier for DCM (Line 23).

2.)  The major mutant gene for DCM is TTN, encoding the giant sarcomere protein TTN. This is not mentioned. Several other genes like DES or RBM20 are also missing. Therefore, the review article could benefit to reference the following book chapter in line 27:

Gerull, Brenda, Sabine Klaassen, and Andreas Brodehl. "The genetic landscape of cardiomyopathies." Genetic causes of cardiac disease (2019): 45-91.

3.)  Please specify that you mean RYR2 (line 40).

4.)  Rephrase sentence in Line 66-68. “… cardiomyocytes derived from human iPSC-CMs.” makes no sense, since you mention cardiomyocytes twice.

5.)  Please add after Line 72 the following review article as a reference. This review article summarizes nicely the iPSC-lines for genetic cardiomyocytes:

Brodehl, A., Ebbinghaus, H., Deutsch, M. A., Gummert, J., Gärtner, A., Ratnavadivel, S., & Milting, H. (2019). Human induced pluripotent stem-cell-derived cardiomyocytes as models for genetic cardiomyopathies. International Journal of Molecular Sciences20(18), 4381.

6.)  Could you present a small schematic figure explaining the GECIs in detail? Only explaining them in the text is hard to follow.

7.)  How are the biosensors localizing to subcellular compartments be designed? (Line 100-103). Please explain in more detl.

8.)  The Figure 2B is unsharp and the subfigures can ne be read. Please update this figure.

9.)  Sometimes the authors write Ca2+ (Line 141) and sometimes calcium (Line 149). Could the authors double check their manuscript for a homogenous writing style.

10.)              Could you provide vector maps for AAV-CAG-RGECO-TnT in the supplements? Is it possible to submit the used plasmids to Addgene?

11.)              Line 305, following: Could you indicate the genotype of the used iPSC-Lines? Which mutations are present in these cell lines?

12.)              Figure 4E: Could you indicate the size of the scale bar?

13.)              Figure 5I: Scale bar size is also not defined.

14.)              Please increase the size of the writing in Figure 6.

15.)              Could you discuss the use of isogeneic controls? I think this is an adequate better control than using wild-type iPSC from a healthy donor. Therefore, I suggest to discuss this point.

However, I am optimistic that the authors can fix these points. In conclusion, I suggest a major revision.

Comments on the Quality of English Language

English quality is fine. 

Author Response

1)  Please add an OMIM identifier for DCM (Line 23).

RESPONSE: Since this disease goes beyond simle genetic causes, we do not think it is relevant here.

2)  The major mutant gene for DCM is TTN, encoding the giant sarcomere protein TTN. This is not mentioned. Several other genes like DES or RBM20 are also missing. Therefore, the review article could benefit to reference the following book chapter in line 27:

Gerull, Brenda, Sabine Klaassen, and Andreas Brodehl. "The genetic landscape of cardiomyopathies." Genetic causes of cardiac disease (2019): 45-91.

RESPONSE: We have referenced several key reviews already.

3)  Please specify that you mean RYR2 (line 40).

RESPONSE: Done

4.)  Rephrase sentence in Line 66-68. “… cardiomyocytes derived from human iPSC-CMs.” makes no sense, since you mention cardiomyocytes twice.

RESPONSE: Corrected. Thanks.

5)  Please add after Line 72 the following review article as a reference. This review article summarizes nicely the iPSC-lines for genetic cardiomyocytes:

Brodehl, A., Ebbinghaus, H., Deutsch, M. A., Gummert, J., Gärtner, A., Ratnavadivel, S., & Milting, H. (2019). Human induced pluripotent stem-cell-derived cardiomyocytes as models for genetic cardiomyopathies. International Journal of Molecular Sciences, 20(18), 4381.

RESPONSE: Again, our focus is beyond one arm of this complicated disease and they reference doesn’t fit with the sentence.

6)  Could you present a small schematic figure explaining the GECIs in detail? Only explaining them in the text is hard to follow.

   RESPONSE: Done. This is a new figure added as Figure 2 in the revised version.

7)  How are the biosensors localizing to subcellular compartments be designed? (Line 100-103). Please explain in more detl.

RESPONSE: These are described in the attached references, beyond the scope of a methods article here. We did describe the biosensor we used in detail.

8.)  The Figure 2B is unsharp and the subfigures can ne be read. Please update this figure.

RESPONSE: The figure was not meant to have data visible, just to give the reader an idea of workflow.

9)  Sometimes the authors write Ca2+ (Line 141) and sometimes calcium (Line 149). Could the authors double check their manuscript for a homogenous writing style.

  RESPONSE: Done

10)              Could you provide vector maps for AAV-CAG-RGECO-TnT in the supplements? Is it possible to submit the used plasmids to Addgene? ADD CREDIT IN ACKNOWLEDGEMENTS?

RESPONSE: We now provide the source and processs in methods and the acknowledgements. The plasmid pShuttle CMV RGECO-TnT was a gift from Matthew Daniels (Addgene plasmid#124643;http://n2t.net/addgene:124643; RRID:Addgene_126463).

11)              Line 305, following: Could you indicate the genotype of the used iPSC-Lines? Which mutations are present in these cell lines?

RESPONSE: Representative calcium transients from hiPSC-CMs derived from a female control subject HID041004, male idiopathic DCM HID041020, and female doxorubicin-induced DCM patient HID041021 were used. This is now stated. The details will appear in a subsequent manuscript.

12)              Figure 4E: Could you indicate the size of the scale bar? 50 micrometres

     RESPONSE: Done

13.)              Figure 5I: Scale bar size is also not defined.

      RESPONSE: Done

14.)              Please increase the size of the writing in Figure 6.

       RESPONSE: Done

15.)              Could you discuss the use of isogeneic controls? I think this is an adequate better control than using wild-type iPSC from a healthy donor. Therefore, I suggest to discuss this point.

RESPONSE: We mention this in the discussion now.

Reviewer 2 Report

Comments and Suggestions for Authors

Good job!

Comments on the Quality of English Language

The quality of english language is good

Author Response

Thanks very much for the kind words!

Reviewer 3 Report

Comments and Suggestions for Authors

Dilated cardiomyopathy is a serious heart condition with limited understanding on its mechanisms. Hawey et al. described a method of transient analysis in iPSC model of DCM using AAV-transduced fluorescent Calcium indicator. There are indeed similar studies using the same or different fluorescent biosensors. That's why, the current study has limited novelty and addition to the literature. Authors should clearly describe what differentiates their method/strategy from previously available ones. I also have some other major and minor comments below:

Major Comments:

Please indicate how long after the differentiation the iPSC-derived cardiomyocytes were harvested for subsequent analyses.

Supplemental S1.1 mentions slight modifications in differentiation method from Ref35. Please briefly explain the methodology of differentiation and indicate differences from published method. Also, I recommend to include some additional data, e.g. % cTnT+ and or a population showing differentiation efficiency and/or quality.

It is not clear what type of statistical analyses were used in the study. Please more specifically indicate appropriate statistical tests and correction methods, e.g. one way ANOVA, used in " 2.3.3. Statistical Analyses" 

How do authors explain the difference in male and female DCM patients in Figure 6? For example; transient frequency is higher male patient while female patient is comparable with control patient. Are the differences in male due to the nature of disease or due to differences in iPSC differentiation? These differences in male patient may indicate a more atrial-like cardiomyocyte features. Authors may compare MLC2A vs. MLC2V ratios to ensure the ratio of differentiated ventricular cells are comparable between patients to eliminate the possibility of variations in differentiation which may have little to do with disease mechanism.

Did authors electrically/optically pace the cells to analyze Ca2+ transients? Although iPSC-CM can spontaneously beat and spark calcium transients, it is best if they are paced to synchronize their beating so that they can be more effectively compared. Otherwise, variability of calcium transients in single cells can make it harder to fairly compare many calcium parameters.

Minor Comments:

Please increase figure resolution.

Scale bar in fluorescent images and text over the scale bars are way too small to see. Please use larger text and thicker scale bar.

Figure 6 has several panels with missing Y-axis labels. Also, the text size is too small to see, and figure resolution needs to be improved.

Comments on the Quality of English Language

Quality of language is great. No need for extensive editing. Some minor typing errors could be fixed.

Author Response

Dilated cardiomyopathy is a serious heart condition with limited understanding on its mechanisms. Hawey et al. described a method of transient analysis in iPSC model of DCM using AAV-transduced fluorescent Calcium indicator. There are indeed similar studies using the same or different fluorescent biosensors. That's why, the current study has limited novelty and addition to the literature. Authors should clearly describe what differentiates their method/strategy from previously available ones. I also have some other major and minor comments below:

RESPONSE: The novelty of the paper is the originlab program – we struggled to find a pipeline that was user friendly, didn’t require knowledge of coding, wasn’t proprietary/required a specific microscope, and that could easily calculate all the parameters we were looking for. The program is useful because it  can be adapted to essentially any workflow, it doesn’t matter how the data is collected as long as intensities can be extracted.

Major Comments:

Please indicate how long after the differentiation the iPSC-derived cardiomyocytes were harvested for subsequent analyses.

RESPONSE Day 28-35 noted in revised version.

Supplemental S1.1 mentions slight modifications in differentiation method from Ref35. Please briefly explain the methodology of differentiation and indicate differences from published method. Also, I recommend to include some additional data, e.g. % cTnT+ and or a population showing differentiation efficiency and/or quality.

RESPONSE: Differentiation of hiPSCs into cardiomyocytes was accomplished via the modulation of Wnt signaling using the previously published GiWi protocol with slight modifications. [35]. Briefly, on day 0 hiPSCs were treated with 12 μM CHIR99021 to activate the WNT signalling pathway. The following day, media was replaced with RPMI1640 supplemented with B27 without insulin. On day 3 the WNT signalling pathway was inhibited using 5 μM IPW2. On day 5 media was replaced with RPMI1640 supplemented with B27 without insulin and on day 7, media was replaced with RPMI1640 supplemented with B27 with insulin. The cell population was purified using 1M lactate to select for hiPSC-CMs. Successful differentiation of hiPSCs into cardiomyocytes (CMs) was confirmed through immunofluorescence and via the expression of the cardiomyocytespecific biosensor RGECO-TnT (Figs. 5 and 6).

We have added this to the supplemental section.

It is not clear what type of statistical analyses were used in the study. Please more specifically indicate appropriate statistical tests and correction methods, e.g. one way ANOVA, used in " 2.3.3. Statistical Analyses" 

RESPONSE: One way ANOVA with a Bonferroni correction- added to methods

How do authors explain the difference in male and female DCM patients in Figure 6? For example; transient frequency is higher male patient while female patient is comparable with control patient. Are the differences in male due to the nature of disease or due to differences in iPSC differentiation? These differences in male patient may indicate a more atrial-like cardiomyocyte features. Authors may compare MLC2A vs. MLC2V ratios to ensure the ratio of differentiated ventricular cells are comparable between patients to eliminate the possibility of variations in differentiation which may have little to do with disease mechanism.

RESPONSE: Thanks for this comment. Investigating these differences isn’t within the scope of this paper. The paper is simply to present a new program that can be used to evaluate and compare calcium transients. We show that using the pipeline we are able to detect differences and further applications can probe differences due to sex and etiologies as well as cell type specific differences. RGECO-TnT could be cloned into a plasmid that has a cell type specific promoter to differentiate atrial vs ventricular calcium signalling in subsequent work.

Did authors electrically/optically pace the cells to analyze Ca2+ transients? Although iPSC-CM can spontaneously beat and spark calcium transients, it is best if they are paced to synchronize their beating so that they can be more effectively compared. Otherwise, variability of calcium transients in single cells can make it harder to fairly compare many calcium parameters.

RESPONSE: Similar to my point above, of course its better to pace the cells but every lab will have their own setup and method to measure calcium, this paper is more to help with the data analysis.

Minor Comments:

Please increase figure resolution.

Scale bar in fluorescent images and text over the scale bars are way too small to see. Please use larger text and thicker scale bar.

Figure 6 has several panels with missing Y-axis labels. Also, the text size is too small to see, and figure resolution needs to be improved.

Corrected- thanks!

Reviewer 4 Report

Comments and Suggestions for Authors

The article entitled „Measuring Single-Cell Calcium Dynamics Using a Myofilament-Localized Optical Biosensor in hiPSC-CMs Derived from DCM Patients” by Hawey al. is an interesting paper. It is particularly valuable to experience a new testing method of calcium uptake and release in cells observed based on the fluorescence signal induced by the calcium transport and thus the cells can be differentiated if they are tilted by dilated cardiomyopathy (DCM) or healthy cells. hiPSC-CMs are also well-konown models for different cardiomyopathy types. Fluorescence spectroscopy is known to be qute sensitive detection method that makes it an excellent choice for the task described in the paper.

Only a minor observation occured, the document entitled „hiPSC MODELING OF INHERITED CARDIOMYOPATHIES” in Curr Treat Options Cardiovasc Med. 2014 Jul; 16(7): 320. is suggested to be refered to. 

Minor revision is suggested for the paper in Cells.

Author Response

Only a minor observation occured, the document entitled „hiPSC MODELING OF INHERITED CARDIOMYOPATHIES” in Curr Treat Options Cardiovasc Med. 2014 Jul; 16(7): 320. is suggested to be refered to. 

RESPONSE: We has already referred to the original paper in this regard: Sun, N., et al., Patient-specific induced pluripotent stem cells as a model for familial dilated cardiomyopathy. Sci Transl Med, 2012. 4(130): p. 130ra47.

Minor revision is suggested for the paper in Cells.

RESPONSE: Thanks for the kind words.

Round 2

Reviewer 1 Report

Comments and Suggestions for Authors

The authors have addressed the majority of points in a sufficient way. In my view this revised manuscript can be published. Congratulations!

Comments on the Quality of English Language

Some minor errors might be double corrected by a language editor.

Reviewer 3 Report

Comments and Suggestions for Authors

Authors clearly addressed majority of my questions. I have no further suggestions.